# The effect of visual sensory interference during multitask obstacle crossing in younger and older adults

HyeYoung Cho[1,2,3], Shirley Rietdyk[1,2] *

1 Department of Health and Kinesiology, Purdue University, West Lafayette, IN, United States of America,
2 Center for Aging and the Life Course, Purdue University, West Lafayette, IN, United States of America,
3 Department of Kinesiology, University of Northern Iowa, Cedar Falls, IA, United States of America

These authors contributed equally to this work.
* srietdyk@purdue.edu

**Data Availability Statement:** The dataset is publicly available at Purdue University Research Repository. https://purr.purdue.edu/publications/4384/1.

## Abstract

When older adults step over obstacles during multitasking, their performance is impaired; the impairment results from central and/or sensory interference. The purpose was to determine if sensory interference alters performance under low levels of cognitive, temporal, and gait demand, and if the change in performance is different for younger versus older adults. Participants included 17 younger adults (20.9±1.9 years) and 14 older adults (69.7±5.4 years). The concurrent task was a single, simple reaction time (RT) task: depress button in response to light cue. The gait task was stepping over an obstacle (8 m walkway) in three conditions: (1) no sensory interference (no RT task), (2) low sensory interference (light cue on obstacle, allowed concurrent foveation of cue and obstacle), or (3) high sensory interference (light cue away from obstacle, prevented concurrent foveation of cue and obstacle). When standing, the light cue location was not relevant (no sensory interference). An interaction (sensory interference by task, p<0.01) indicated that RT was longer for high sensory interference during walking, but RT was not altered for standing, confirming that sensory interference increased RT during obstacle approach. An interaction (sensory interference by age, p<0.01) was observed for foot placement before the obstacle: With high sensory interference, younger adults placed the trail foot closer to the obstacle while older adults placed it farther back from the obstacle. The change increases the likelihood of tripping with the trail foot for younger adults, but with the lead limb for older adults. Recovery from a lead limb trip is more difficult due to shorter time for corrective actions. Overall, visual sensory interference impaired both RT and gait behavior with low levels of multitask demand. Changes in foot placement increased trip risk for both ages, but for different limbs, reducing the likelihood of balance recovery in older adults.

## Introduction

Community mobility requires the ability to identify and avoid hazards while walking [1]; this skill is critical to maintaining independence in later life [2]. Concurrent tasks are often

**Funding:** The author(s) received no specific funding for this work.

**Competing interests:** The authors have declared that no competing interests exist.

performed while walking, called dual-task or multitask, such as avoiding pedestrians, talking to a partner, and/or attending a traffic signal. The ability to multitask is impaired in older adults due to deteriorations of sensory, motor, and/or cognitive systems [3–6]. Multitask performance can provide a window into the mechanisms underlying age-related impairments that lead to falls [7]. For example, while visuospatial working memory (VSWM) is an important component for obstacle crossing, older adults' ability to store and use an obstacle representation via VSWM is not different from younger adults [8], indicating that mechanisms other than VSWM are responsible for the age-related impairments in locomotor multitasks.

The visual system is critical for locomotion as it provides information of upcoming hazards, allowing proactive modifications to safely avoid hazards [9–11]. Sensory interference, also known as structural interference, results when the same perceptual modality is used for more than one task, such as using vision to complete both the cognitive task and the gait task [12]. Visual sensory interference is commonplace during daily activities; for example, using vision to identify (1) the color of the traffic signal and (2) the location of the sidewalk curb. Sensory interference is distinct from central interference, also known as capacity interference, which refers to tasks competing for a limited amount of cognitive resources [12].

Sensory interference is affected by age and locomotor task. In younger adults, sensory interference did not alter foot targeting performance [13], but did alter performance in obstacle crossing [14, 15] and stair descent [16]. These changes are consistent with the idea that more difficult tasks result in greater multitask impairment [17, 18]. In older adults, sensory interference increased foot placement error during foot targeting [13] and increased obstacle contacts and slowed gait on an obstacle course [19]. Other studies have demonstrated impaired performance in older adults during multitask obstacle crossing [20–25], but these studies manipulated sensory and/or central interference, such that the role of sensory interference alone cannot be determined. Since tripping is one of the most common causes of falls, and trip-related injuries increase with age [26–32], it is important to further examine the effect of sensory interference during obstacle crossing. For example, gaze that is diverted momentarily while crossing a street is a common occurrence. The discrete nature of a brief gaze diversion may not impair performance to the same extent as when gaze is diverted repeatedly or for longer periods, such as in [19].

In previous studies, the nature and complexity of the gait task and the concurrent tasks have varied widely and it is important to specify the task complexity [7, 14]. A framework to identify relative complexity of locomotor multitasks includes difficulty of the gait task, difficulty of the concurrent (non-gait) task, duration of the non-gait task, and magnitude of the sensory interference (Fig 1; adapted from [14]). The gait task demands vary as a function of the number and characteristics of hazards in the walkway (e.g., size, contrast). For the non-gait task, cognitive demands are lower for a simple reaction time (RT) task versus a choice RT task, and also appear lower for external interfering factors (e.g., RT tasks) versus internal interfering factors (e.g., mental tracking tasks) [33]. Temporal demands are lower for a single discrete task versus more continuous tasks (e.g., listening to a story).

The current study focuses on the less difficult side of the framework (vertical arrows in Fig 1). Since higher challenges are associated with greater performance impairment [17, 34], a change in performance with lower-challenge tasks will provide strong evidence of the impairing effect of sensory interference [14]. Sensory interference was created with the location of the RT light cue. The cue was located either (1) on the obstacle (low sensory interference since obstacle and RT cue can be foveated concurrently) or (2) away from the obstacle (high sensory interference as the obstacle and RT cue cannot be foveated concurrently) (Fig 2A). Further, to delineate that any change in RT was due to sensory interference, we included a standing task

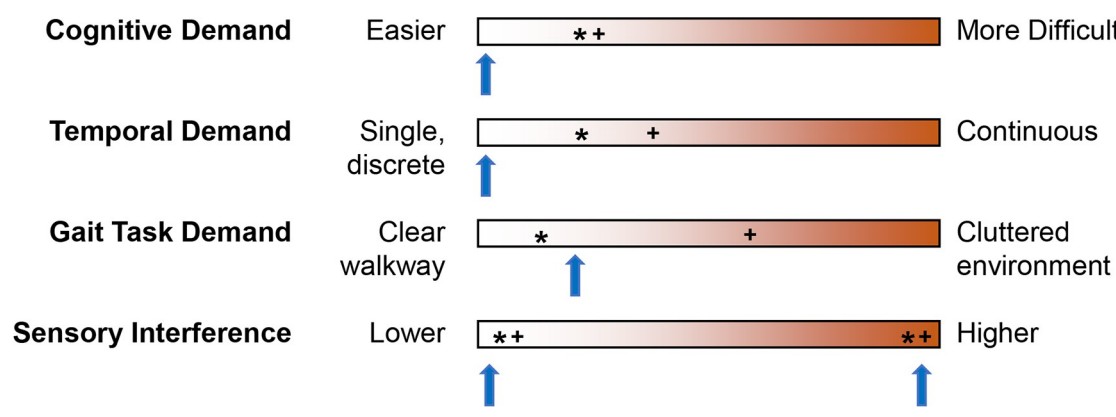

| Study | Cognitive Task | Temporal Component of Cognitive Task | Gait Task | Age Groups |
|---|---|---|---|---|
| Current study | Simple RT – push button when light cue activated | Cue lighted once during approach to obstacle | Walk and step over single, stationary obstacle; 8 m walkway | 17 YA 14 OA |
| * Berg & Murdock, 2011 | Verbally identify one letter on monitor | Single letter presented on monitor every 1-2 s | Walk and step on single, stationary target (1.2 m wide by 0.29 m deep); 11.6 m walkway | 20 YA 20 OA |
| + Menant et al., 2009 | (1) Verbally identify three letters on monitor at eye level at end of walkway and (2) verbally identify suit of playing card on side of walkway | Three letters presented every 3.5 s, playing card viewed two times | Walk and step over 21 stationary obstacles, including 14 low contrast obstacles; 14.5 m walkway | 13 OA |

YA = younger adults
OA = older adults

**Fig 1.** Top panel: Framework indicating demands examined with multitask locomotor research. Relatively lower demands are on the left, with increasing demands to the right in red. Demands of the current study are demonstrated with blue vertical arrows (note that the arrow position demonstrates relative differences across conditions and studies, and it not intended to reflect absolute differences or locations on the spectrum). Two examples of previous sensory interference research are also indicated: (1) * foot target task [13] and (2) + obstacle task [19]. Bottom panel: Information about each study. All studies compared low versus high visual sensory interference. The current study focuses on the less difficult side of the framework, to determine if visual sensory interference alters performance under low levels of cognitive, temporal, and gait demand.

(Fig 2B). When standing, the location of the RT cue does not create sensory interference since obstacle information is not relevant to standing.

The purpose of this study was to determine if visual sensory interference alters performance under low levels of cognitive, temporal, and gait demand, and if the change in performance is different for younger versus older adults. We hypothesized that RT will be longer with high versus low sensory interference for the gait task but not the standing task (H1: interaction effect of sensory interference by task). We hypothesized that older adults will have a greater increase in RT (H2: main effect of age or an interaction with age). We hypothesized that gait measures will be affected more with high versus low sensory interference for both age groups,

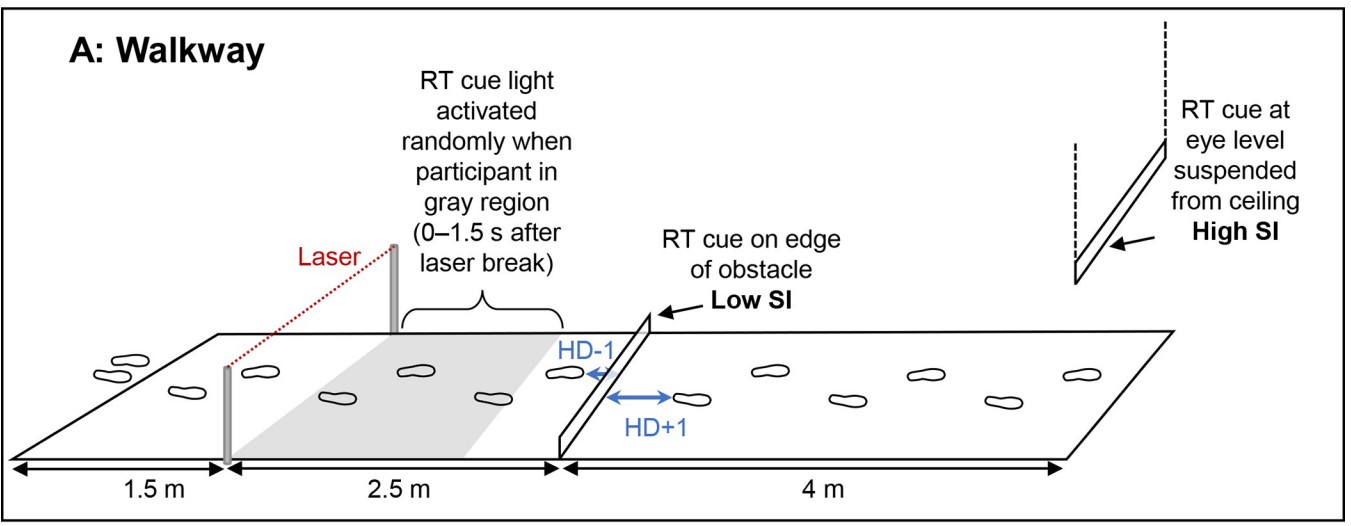

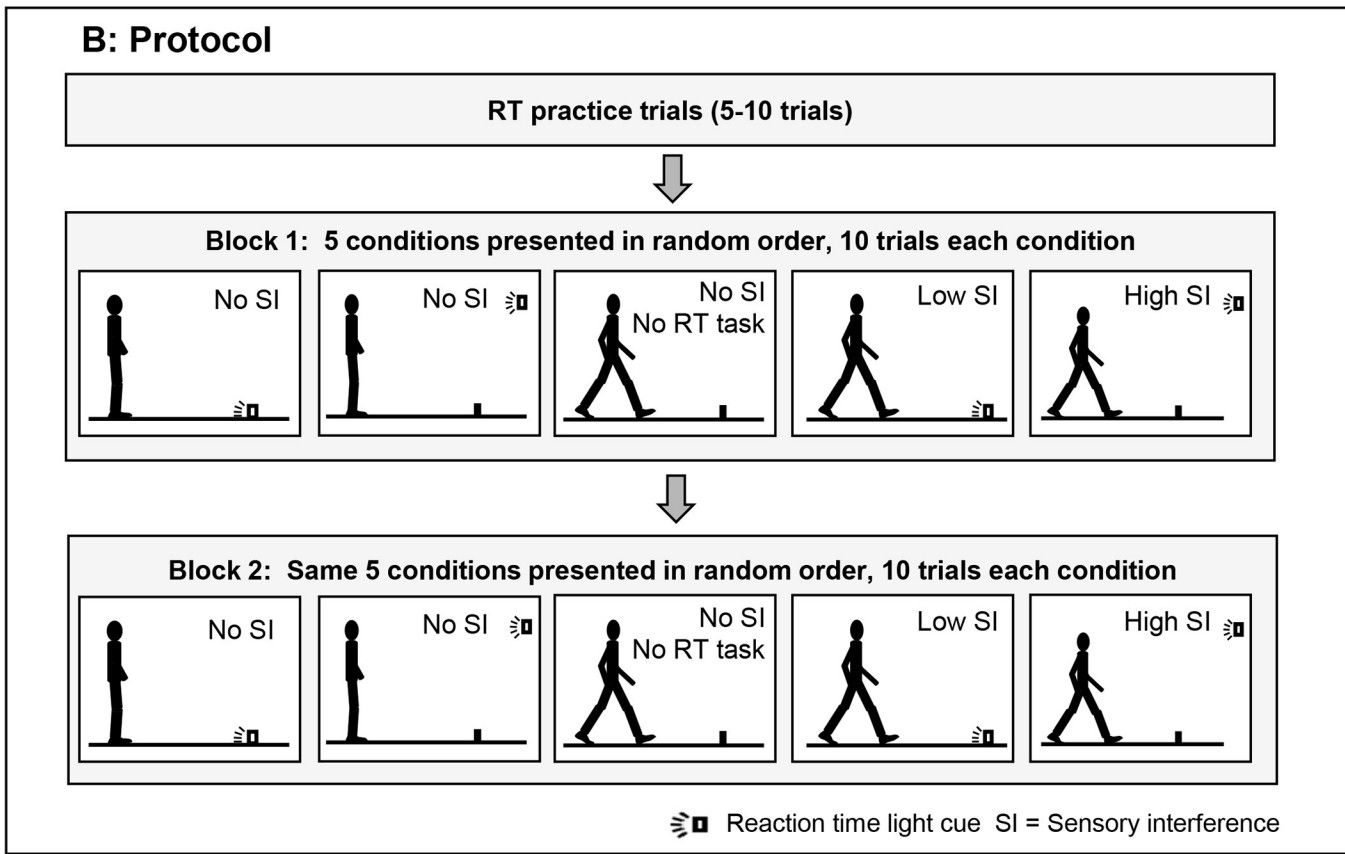

**Fig 2.** (A) Line drawing of the walkway, obstacle, and RT cue. In the walking conditions, the RT cue was triggered 0–1.5 s (corresponds to gray region) after the participant broke the laser beam (positioned at hip height). The RT cue appeared once during approach, either on the obstacle (low sensory interference (SI)) or at eye-level (high SI). Participants pressed a hand-held remote when the RT light cue turned on. HD-1 and HD+1 represent the horizontal distance from the obstacle to the trail toe and the lead heel, respectively. (B) The block-randomized protocol. In the standing conditions, participants stood 1.5 m in front of the obstacle, and the RT cue appeared on either the obstacle or at eye-level. For standing trials, there was no SI regardless of light cue location. SI = sensory interference.

and older adults would have a greater change in gait measures (H3: interaction effect of sensory interference by age). Acceptance of these hypotheses will demonstrate that sensory interference impaired the performance of the multitask even when the concurrent task demands are relatively low, and sensory interference had a greater effect on older adults. Further, since multitask gait performance can improve with short-term training [35, 36], we also hypothesized that performance will improve in the second block of trials (H4: main effect of block or an interaction with block).

## Methods

### Participants

Seventeen younger adults (3 males, 20.9 ± 1.9 years, 1.66 ± 0.71 m, 67.4 ± 11.5 kg) and fourteen older adults (4 males, 69.7 ± 5.4 years, 1.70 ± 0.72 m, 79.7 ± 21.9 kg) participated in this study; they were recruited between the 16th of March and the 2nd of November in 2017. The results for the older adult participants have not been published. The results for the younger adult participants were previously published [14], and are included here as the comparison group for older adults. The older adults lived independently (i.e., they did not require assistance with daily activities). Exclusion criteria included neuromuscular or orthopedic disorders that affected walking. The study was approved by the University Institutional Review Board and all participants provided written informed consent.

### Experimental set up

Full-body kinematic data were collected by 8-camera motion capture system (Vicon Vero, Oxford, UK) at 250 Hz. Marker clusters were placed on 13 segments: head, thorax, lower back, upper arms, forearms, thighs, shanks, and feet. Joint centers and anatomical points (including first metatarsal head and calcaneus) were digitized. Participants wore comfortable walking or athletic shoes; they were asked to wear the shoes they habitually wear when walking.

An obstacle was placed in the middle of an 8 m walkway; obstacle width and depth were 100 cm and 0.8 cm. The obstacle height was either 21 cm or 23.5 cm; obstacle height was dependent on participant's leg length as follows: leg length <86 cm: 21 cm obstacle and leg length ≥86 cm: 23.5 cm obstacle. The two obstacle heights provided a similar percentage of obstacle height to leg length: mean obstacle height was 24.9 ± 1.2% of leg length. Two obstacles of fixed height were used (rather than multiple obstacles with height as a fixed percent of leg length) so that only two light cues were needed for mounting on the obstacle. The obstacle was designed to fall forward if contacted, like a hurdle, to decrease fall-risk. Thus, participants were not harnessed. No one fell in this protocol or in our previous studies with this obstacle design (e.g., [37, 38]).

The stimulus for the reaction time task was LED strip lights shaped into two rectangles. The rectangle was the same height and width as the obstacle; one rectangle was attached to the edge of the obstacle, and a second identical rectangle was suspended at eye level at the end of the walkway. When the RT cue was on the obstacle, the obstacle and RT cue could be foveated concurrently (assumed to be "low sensory interference"). When the RT on was at eye height, the obstacle and RT could not be foveated concurrently (assumed to be "high sensory interference"). The RT cue was triggered by a laser beam located 2.5 m before the obstacle and 1 m in height (Fig 2A); the RT cue was randomly activated between 0 and 1.5 seconds after the laser was broken (gray region in Fig 2A). RT cue duration was 300 ms.

## Procedure

Before data collection, practice with the RT task while standing was provided and continued until the participant was confidently completing the RT task (5–10 trials). Participants completed five conditions (Fig 2B): 1) standing with RT task on the obstacle (no sensory interference), 2) standing with RT task at eye level away from the obstacle (no sensory interference), 3) obstacle crossing baseline without RT task (no sensory interference), 4) obstacle crossing with RT task on the obstacle (low sensory interference), and 5) obstacle crossing with RT task at eye level away from the obstacle (high sensory interference). The RT task occurred once during the approach phase to the obstacle (Fig 2B). In the standing condition, participants stood 1.5 m before the obstacle (distance selected for similarity to obstacle crossing conditions) and the computer randomly activated the light cue (10 trials were completed in 90 seconds). Participants were informed where the RT cue would appear at the beginning of each condition. Participants carried a wireless clicker and were instructed to depress it as soon as possible when the light cue was activated. Trials were collected in two blocks (Fig 2B). In the first block, 10 trials of each of the five conditions were collected, and conditions were block-randomized. In the second block, 10 more trials of each condition were collected, and conditions were also block-randomized (different order than first block). Two blocks were included to determine if performance changed with repeated trials. A total of 100 trials were completed for each participant (60 gait trials and 40 standing trials); the standing/walking protocol lasted about one hour. Participants took a required 5-minute break between the two blocks. Participants were asked if they needed a longer break; no longer breaks were requested.

## Data analysis

RT, the time between light cue activation and pressing the wireless clicker, was analyzed. Out of 3100 total trials, 15 trials were excluded as follows: 14 trials (7 younger adults, 7 older adults) due to potential anticipatory reaction (RT<120 ms), and 1 trial (1 older adult) due to lack of attention (RT>1100ms) [39].

Three-dimension full body kinematic data was analyzed with Motion Monitor (IST Inc., IL, USA) and Matlab R2014a (Mathworks Inc., MA, USA). A fourth-order zero-lag low-pass Butterworth filter was used for kinematic data with 7 Hz cut-off frequency [40]. Trail foot placement (HD-1) was the horizontal distance between the trail toe and the obstacle and lead foot placement (HD+1) was the horizontal distance between the lead heel to obstacle horizontal distance after crossing the obstacle (Fig 2A). The lead and trail foot clearances were determined by the minimum value of toe and heel vertical clearance over the obstacle [41–43]. Gait speed was the mean of the anterior-posterior center of mass (COM) velocity in three steps: one-step before the obstacle crossing, the lead step crossing the obstacle, and the trail step crossing the obstacle.

## Statistical analysis

Generalized linear mixed model ANOVA was used for this study with SAS 9. 3 (SAS Institute, Cary N.C.). The between-subject factor was age and the within-subject factors were sensory interference and block. A Kenward-Roger correction was applied to account for different number of trials due to anticipatory RT (see above) and trials with an obstacle contact (Table 1). RT was analyzed with a four-way ANOVA (age, task, sensory interference, and block). Gait variables were foot clearance, foot placement for trail and lead limbs (HD-1 and HD+1), and gait speed. Gait measurements were analyzed with three-way ANOVAs (age, sensory interference, block). Due to the large number of dependent variables, significance level

**Table 1. Number of obstacle contacts.** The percent value in the last column is percent of all trials.

| Age | Condition | Block 1 | Block 2 | Total |
|---|---|---|---|---|
| Younger adult (N = 17) | Gait baseline (no RT task) | 2 | 1 | 3 (0.9%) |
| | RT cue on obstacle | 1 | 1 | 2 (0.6%) |
| | RT cue away from the obstacle | 4 | 2 | 6 (1.8%) |
| | Three conditions combined | 7 | 4 | 11 (1.1%) |
| Older adult (N = 14) | Gait baseline (no RT task) | 0 | 3 | 3 (1.1%) |
| | RT cue on obstacle | 0 | 3 | 3 (1.1%) |
| | RT cue away from the obstacle | 1 | 2 | 3 (1.1%) |
| | Three conditions combined | 1 | 8 | 9 (1.1%) |

was set at $p \leq 0.01$ to reduce the likelihood of false positive errors. Only significant effects are reported.

## Results

### Reaction time

RT was significantly affected by the two-way interaction of task by sensory interference ($F_{(1, 57.8)}$ = 14.63, p<0.001; Fig 3A). Post hoc analysis revealed that RT in the standing task was not affected by the sensory interference, but RT in the gait task was longer with high versus low sensory interference for both younger and older adults (see asterisks (*) in Fig 3A).

RT was also significantly affected by the three-way interaction of age by task by block ($F_{(1, 29)}$ = 9.88, p = 0.004; Fig 3B). Post hoc analysis revealed that in block 1, RT during gait in older adults was significantly longer than RT during standing in both age groups. In block 2, RT during gait in older adults was not different from RT during standing in either group. In younger adults, RT was not affected by any combination of task and block.

### Gait measures

**Obstacle contacts.** No statistical analyses were conducted on obstacle contacts because of the nature of the data; descriptive results are provided. Twenty obstacle contacts were observed as follows: 11 contacts by eight younger adults (47% of younger adult participants) and nine contacts by five older adults (36%) (Table 1). Two younger adults (12%) and two older adults (14%) contacted the obstacle more than once. For younger adults, 64% and 36% of contacts occurred in the first and second block, respectively. For older adults, 11% and 89% of contacts occurred in the first and second block, respectively (Table 1). The contact limb was the lead limb for 18% and 44% of the contacts, for younger and older adults, respectively.

**Foot placement.** Foot placement before the obstacle (HD-1) was significantly affected by an interaction of age by sensory interference ($F_{(2, 58.1)}$ = 7.70, p<0.001; Fig 4A). Post-hoc analysis revealed that, for younger adults, HD-1 was closer to the obstacle for high versus low sensory interference. For older adults, HD-1 was farther away with low sensory interference relative to gait baseline. Note that a higher value of HD-1 is considered safer because the person is less likely to trip on the obstacle [41, 44]. Foot placement after the obstacle (HD+1) was significantly affected by the main effect of age ($F_{(1, 29)}$ = 7.20, p = 0.01; Fig 4B), where younger adults placed their lead foot farther from the obstacle (after crossing the obstacle) than older adults.

**Gait speed.** Gait speed ANOVA results were similar for the three observed steps: one-step before the obstacle, the lead step across the obstacle, and the trail step across the obstacle. For all three steps, gait speed was significantly affected by the main effects of age ($F_{(1,29)} \geq 18.6$,

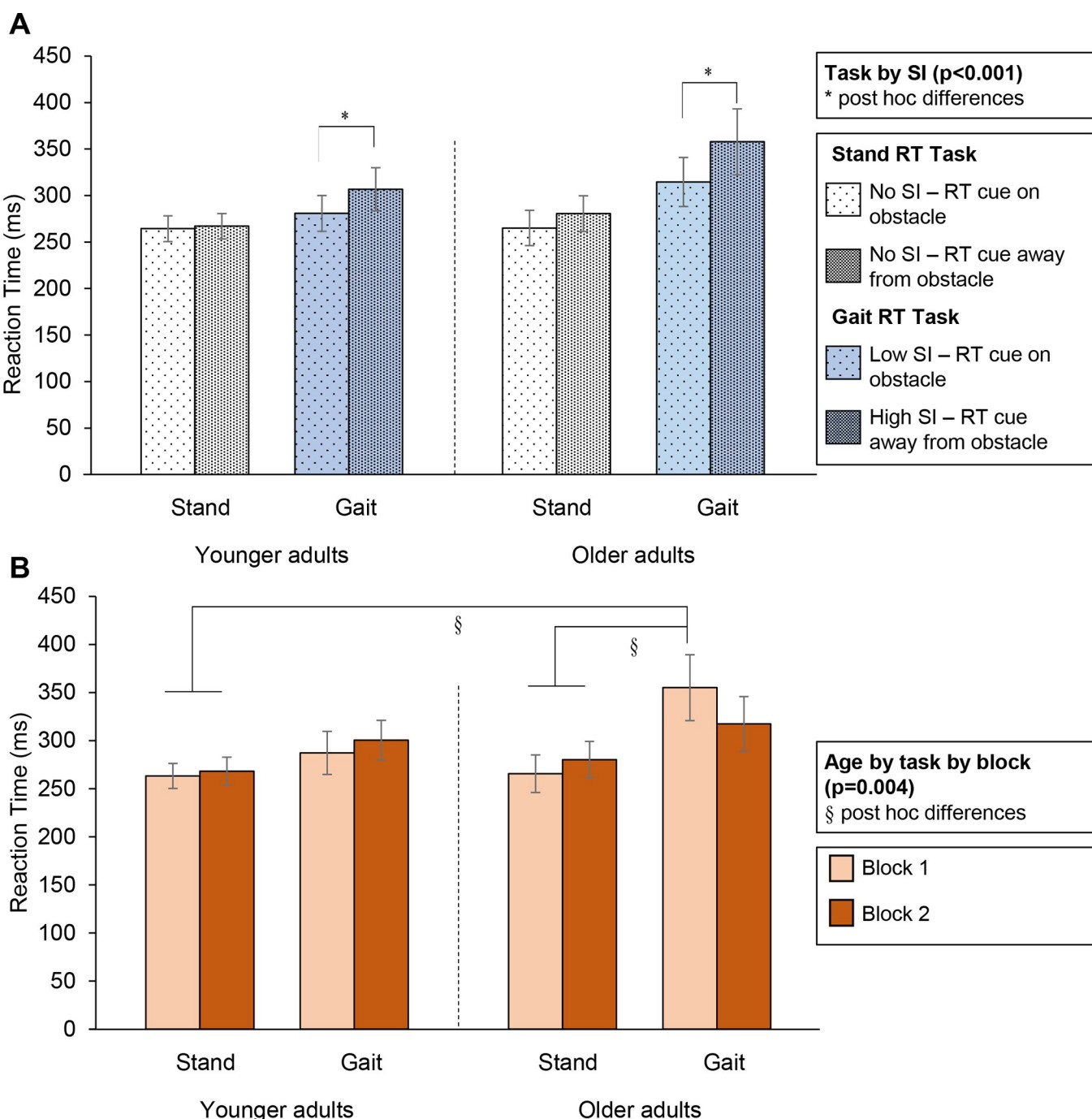

**Fig 3.** (A) Mean reaction time as a function of task and sensory interference. An interaction of sensory interference by task was observed (p<0.001); post hoc differences noted with *. (B) Mean reaction time as a function of age, task, and block. An interaction of age by task by block was observed (p = 0.004); post hoc differences noted with §. Error bars represent standard error. SI = sensory interference.

p<0.001; Fig 5) and sensory interference (F(2,50) ≥ 8.2, p<0.001). Regarding the age effect, for all three steps, gait speed was slower in older than younger adults. Regarding the sensory interference effect, for all three steps, gait speed was slower in the gait baseline condition (no sensory interference) versus both low and high sensory interference (speed was not different for high versus low sensory interference).

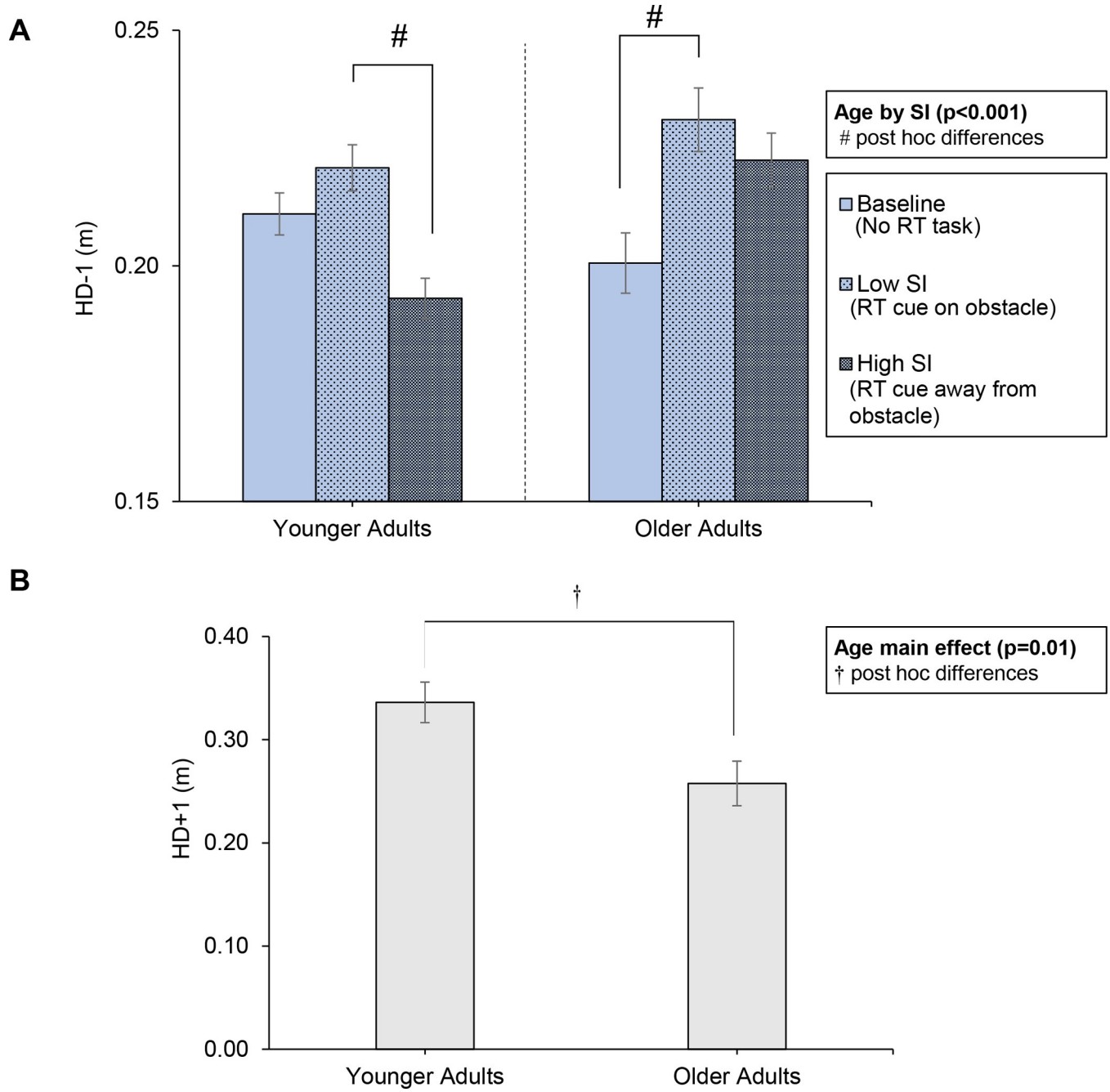

**Fig 4.** (A) Mean trail foot placement before the obstacle (HD-1) as a function of sensory interference in younger and older adults. An interaction of sensory interference by age was observed; post hoc differences noted with #. (B) Mean lead foot placement after the obstacle (HD+1) for young and older adults. An age main effect was observed (p = 0.01) and noted with †. Error bars represent standard error. SI = sensory interference.

**Foot clearance.** Lead foot clearance was not significantly different for any main or interaction effects. Trail foot clearance was affected only by the main effect of block (F(1, 29) = 11.21, p = 0.002): trail foot clearance was higher in block 1 than block 2 (0.15 ± 0.06 m and 0.13 ± 0.06 m).

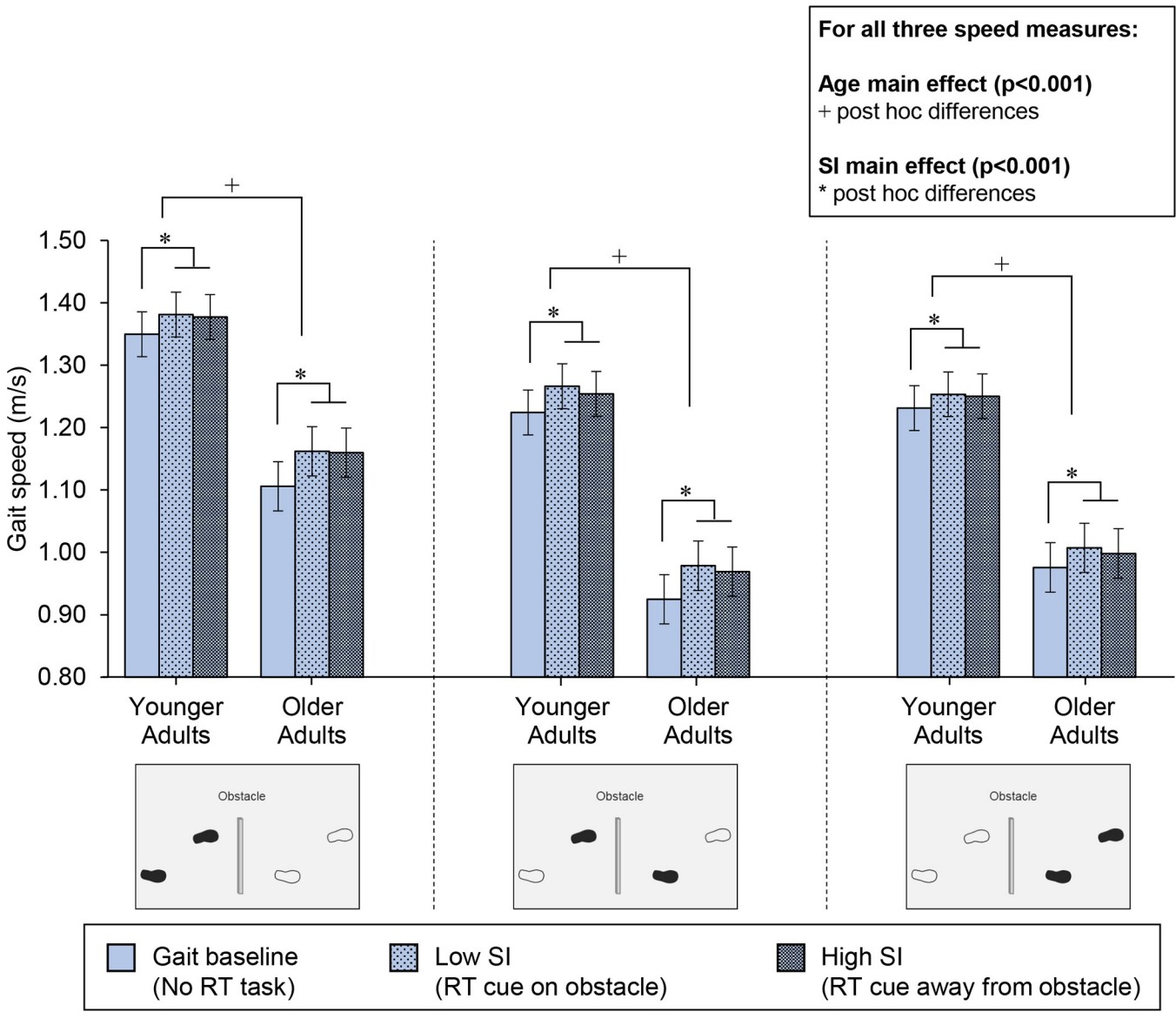

**Fig 5. Mean gait speed as a function of step, sensory interference, and age group.** The footprint figures indicate the step where speed was calculated for each of the three steps (one-step before the obstacle crossing, the lead step across the obstacle, and the trail step after the obstacle). For each of the three steps, two main effects were observed: (1) age (p<0.001); post hoc differences noted with +, and (2) sensory interference (p<0.001); post hoc differences noted with *. Error bars represent standard error. SI = sensory interference.

## Discussion

The purpose of this study was to determine if visual sensory interference alters performance under low levels of cognitive, temporal, and gait demand, and if the change in performance is different for younger versus older adults. Current knowledge was extended by establishing that with a simple, discrete RT task, visual sensory interference (1) impaired RT performance to a similar extent for both younger and older adults, (2) affected foot placement differently for younger and older adults, and (3) did not affect foot clearance for either age group. These findings support the contention that vision plays a critical role during the approach to an obstacle for both younger and older adults. Furthermore, RT during the gait task improved with

repeated exposures, but only for the older adults. These observations have implications for fall-prevention training and the built environment.

Gaze diversion away from the obstacle increased sensory interference, leading to longer RT and modification of gait behavior in both younger and older adults. In the current study design, the cognitive demands of the concurrent, simple RT task were held constant while manipulating sensory interference by diverting gaze toward the obstacle versus away from the obstacle. Thus, any changes in performance resulted from sensory interference of the visual system, and not from central interference. During approach to the obstacle, visual information of obstacle position and size is gathered [11, 45, 46]; this information is relevant for the obstacle crossing task, but not the standing task. The observation that sensory interference altered RT for the gait task but not the standing task supports and strengthens the interpretation that performance impairment was due to sensory interference.

In a different gait task, locomotor targeting, sensory interference impaired foot targeting accuracy in older adults, but not in younger adults [13], whereas we found foot placement differences in both age groups. The lack of sensory interference in younger adults in Berg & Murdock [13] could result from one or more of the following differences: locomotor task (foot targeting vs. obstacle avoidance), RT task (verbally identifying letter on monitor vs. pressing hand-held button when light cue was on), the temporal nature of the cognitive task (multiple discrete tasks vs. single discrete task), consequence for failure (low consequence due to harness vs. higher consequence due to no harness and possibility of contacting the obstacle), and location of gaze diversion in low sensory interference condition (gaze diverted *toward* foot placement target vs. gaze diverted *directly* to the obstacle). Furthermore, the cognitive task performance was not measured in Berg & Murdock [13], and it is possible that younger adults may have responded slower to the concurrent task during locomotor targeting but kept their gait similar, prioritizing the foot targeting gait task over the cognitive task. For obstacle crossing tasks, younger adults demonstrated sensory interference with a visual Stroop task [15], and older adults walked slower and contacted more obstacles when their vision was altered with multifocal lenses relative to single lenses [19]. In the latter study, the multifocal lenses increased sensory interference in a continuous manner, the 14.5 m walkway contained 21 obstacles (about one obstacle for every 0.7 m), two-thirds of the obstacles were low contrast, and the secondary task (verbally identifying letters or the suit of a playing card) was repeated while walking [19]. In the current study, we extend these observations with a more challenging environment and task [19] to demonstrate that sensory interference also altered performance with a single, high-contrast obstacle, and a discrete simple RT task (Fig 1), for both younger and older adults.

The longer RT with high sensory interference may indicate that the participants were switching attention between the obstacle in the walkway and the light cue when it was away from the obstacle. Alternatively, increased RT may indicate that participants adopted a gaze strategy observed in sports, called *gaze anchor*: foveating on one area of interest (e.g., obstacle) and relying on peripheral vision to monitor another area (e.g., light cue location in upper visual field) [47]. Since RT is longer when the stimulus is in the peripheral versus central visual field [48, 49], increased RT may have resulted from monitoring the light cue location with peripheral vision. Thus, gaze strategies during sensory interference with multitasks should be examined in future studies to identify putative factors for interventions. Overall, the impaired RT performance with a relatively easy concurrent task extends and emphasizes the contribution of vision when approaching hazards in the environment for both younger and older adults [9, 11, 45, 46, 50–52].

We did not accept our hypothesis that older adults would have a longer RT than younger adults and/or a greater increase in RT with sensory interference than younger adults (H2). We

based this hypothesis on the longer reaction times observed during multitasking in older adults from previous research (e.g., [25, 53]). However, we note that a recent publication regarding multitask obstacle crossing also demonstrated no age-related main or interaction effects [8]. Similarly, our findings indicate that the impairing effect of gaze diversion on RT, with a simple, visual RT task, was independent of age. The older adults appear to maintain their ability to quickly detect light cue onset while walking despite age-related changes. This observation supports the use of visual cues to highlight hazards in the built environment (e.g., [54–58]), as our results indicate that these cues can be detected and acted on with similar latencies for younger and older adults. We note that a recent review reported that age-related differences in locomotor multitasks were most likely to be observed when the concurrent task had a visual component (see review in [59]), but the tasks included in the review typically included identification of a letter (with 26 options), rather than light cue onset. Thus, any cues provided in the built environment should not include text (such as a sign) but should identify an upcoming hazard through color or lights.

While age-related differences were not observed in RT, they were observed in foot placement measures (HD-1 and HD+1). Changes in foot placement measures reflect changes in visual information regarding obstacle *position* [45, 46, 60, 61]. With low sensory interference relative to no sensory interference (gait baseline), young adults had adequate obstacle position information, indicating that they were able to gather the obstacle position information when they could concurrently foveate the obstacle and light cue in the low sensory interference condition. However, with high sensory interference, younger adults placed the trail foot (HD-1) closer to the obstacle (Fig 4A), likely reflecting a change in obstacle position information. Closer trail foot placement increases the risk of obstacle contact with the trail foot due to reduced time and space to elevate the limb [41, 44]. Conversely, older adults did not have a difference in trail foot placement for high versus low sensory interference. In fact, the foot placement was placed farther back for low sensory interference condition relative to no sensory interference (gait baseline), and foot placement for high sensory interference demonstrated the same trend relative to no sensory interference (p = 0.052) (Fig 4A). A farther trail foot placement will reduce trail limb trip-risk [41, 44], but will increase lead limb trip-risk [62]. Although only a small number of contacts were observed, the contact results support increased lead limb trip risk in older adults: The contact limb was the lead limb for 18% of younger adult contacts and 44% of older adult contacts. The change in foot placement behavior–placing the trail foot farther back behind the obstacle–has also been observed when the lower visual field is obstructed with goggles [11, 46, 61]. Thus, a possible interpretation of the change in foot placement in older adults is that *obstacle position information* is blocked or impaired by the RT task, regardless of the location of the task. Farther foot placement may have occurred if older adults adopted a *gaze anchor* strategy where they foveated on the light cue and maintained the obstacle in peripheral vision, impairing foot position information. Note that in the community, warning signs are often located above hazards (e.g., airplane passenger boarding bridge, or jetway, have overhead signs warning of trip hazards where the jetway floor changes elevation). Presumably warning signs are placed overhead due to the typical forward gaze. However, the findings here indicate that the warning sign should not divert gaze from the hazard.

Overall, the opposite effect of sensory interference on foot placement is interesting: closer trail foot placement for younger adults and farther trail foot placement for older adults. This observation may be especially relevant for stair-related falls, which are common in both younger and older adults, as stairs require accurate foot placement [63, 64]. The role of sensory interference should be considered when developing interventions to counter the high rate of falls and fall-related injuries observed in younger adults [28, 30, 32, 63, 65, 66]. When younger

adults are visually distracted by other tasks, such as using electronic devices, they may be at greater risk of impaired locomotor behavior.

We point out that sensory interference altered foot placement, but not foot clearance. As noted above, foot placement is associated with obstacle *position* information, while foot clearance is associated with obstacle *height* information [14, 45]. The light cue was activated in the region where *position* information is typically gathered [15, 45]. Since only placement was altered, and not height, the findings support the contention that position information must be gathered at a specific location during the approach, but there is more flexibility in when height information can be gathered [45]. An alternate explanation is that the low-risk/collapsible obstacle resulted in complacency during this lab task and participants did not adjust their foot clearance. However, it seems unlikely that complacency would not affect foot clearance but would affect foot placement. In addition, in previous studies with collapsible obstacles, foot clearance has been altered by visual manipulations [67, 68]. Therefore, the discrete, simple RT task apparently affected the ability to gather obstacle position information but did not alter the ability to gather height information.

A common observation during gait multitasking is a decrease in gait speed [69–71], but here we observed faster gait speed in both age groups when multitasking. The increase in speed may have resulted from the following strategy: Participants walked slower in the first 2–3 m of the walkway to allow the light cue to activate well before they reached the obstacle. After completing the RT task, they increased speed to compensate for the slower approach speed. This strategy has been observed during virtual reality: participants slowed down when obstacle information was withheld and then increased speed after the obstacle information was provided [45]. In the current protocol, the light-cue-activation region was outside the motion capture volume, so we could not confirm this strategy.

Short-term multitask training has been shown to improve both gait and RT measures in older adults [36] and in people with Parkinson's disease [35]. The shorter RT for older adults for the second block of trials (Fig 3B) is consistent with these studies, and adds to the growing body of research that multitask training is an effective approach to improve performance in various populations [35, 36, 72–77]. In the current study, the participants had practice trials before the protocol began, and then repeated the RT task 80 times: 40 times while standing and 40 times while approaching an obstacle. RT did not decrease in younger adults, which may indicate that they were already at their minimum RT for this task. The RT decrease in older adults appears to reflect improved performance due to the repeated trials. However, 89% of the obstacle contacts for older adults occurred in the second block (versus 36% for the younger adults; Table 1). In younger adults, there is no evidence to support an association between fatigue and contacts during repeated walking trials [41], consistent with the relatively even distribution of contacts across the two blocks. In older adults, the skewed distribution may result from greater fatigue [38] and/or inattention in the second block. The fatigue most likely resulted from repeatedly stepping over the obstacle [38]. Inattention may also result from fatigue, also leading to obstacle contacts. However, fatigue and inattention are inconsistent with the faster RT observed in block 2 for older adults. An alternate explanation is that, in the second block, the older adults may have shifted attention away from the gait task and toward the RT task. It is also possible that both factors (fatigue and shifted attention) are contributing to the change in RT and obstacle contacts. The study design adopted here does not allow us to differentiate between fatigue and shifting attention. Given these observations, future studies should quantify self-reported fatigue before, during, and after gait protocols to clarify the role of fatigue in locomotor research in various populations [38, 78]. The other parameter that changed in the second block was a decrease in trail foot clearance; the decrease in clearance was not different between age groups. A systematic decrease in foot clearance has been

observed previously when people step over the same obstacle repeatedly [41, 50]; the underlying mechanism of this behavior is not yet understood, but it is consistent with drifts observed in finger forces (e.g., [79]).

The changes in performance during a locomotor multitask highlight potential avenues for fall prevention interventions. While multitask intervention studies have focused on manipulation of task difficulty [75, 80, 81], interventions focused on sensory interference are not as prevalent. The results here–that visual sensory interference resulted in performance impairment–indicates that gaze behavior training may be a viable intervention for preventing trips during activities where gaze is diverted away from upcoming hazards. Gaze behavior training–training people when and where to look, including duration and frequency–has shown improvement in stepping performance including a decrease in foot placement errors and obstacle contacts [51, 82–85]. When multitask training is considered, care must be taken to ensure that older adults do not prioritize the cognitive task over the gait task, as was apparently observed in this study (decreased RT but increased obstacle contacts in the second block).

Furthermore, the built environment should be modified to ensure that gaze is not diverted away from critical features of the environment. Note that we are not implying that people should not scan the environment, but rather that hazard cues are readily visible within the environment. For example, current crosswalk signals are relatively small and because they are at eye height, they divert gaze away from the sidewalk and curb. New crosswalk signals where the entire vertical post lights up and/or crossing signals embedded in the crosswalk will reduce gaze diversion during road crossing. These approaches would supplement other environmental modifications, such as adding the horizontal-vertical illusion or tread edge highlighters on stairs, that have been shown to increase toe clearance and reduce the risk of trips and falls [54–58].

The study contains limitations. First, we assumed that changing the location of the RT cue would alter sensory interference, consistent with others (e.g., [13]). It is also possible that changes in behavior reflect alternate mechanisms, such as differences in visual acuity for central versus peripheral vision [86]. When the RT cue was on the obstacle, the obstacle was likely viewed in central vision, and when the RT cue was at eye level, the obstacle was likely viewed in peripheral vision. Central vision has greater acuity than peripheral vision [86]. Therefore, gait behavior differences may have resulted from differences in visual acuity rather than sensory interference. Future studies should be designed to further differentiate across potential mechanisms. Second, we did not measure gaze behavior, so it is unknown how the frequency and duration of fixations on the obstacle, walkway, and light cue were modified as a function of age and sensory interference. As noted above, gaze strategies will likely provide insight regarding the underlying mechanisms of impaired performance. Third, cognitive ability affects multitask performance [87–89], but we did not measure cognitive ability in this study. Given that RT was not different for younger versus older adults, it appears that cognitive ability in our older adults was adequate. Fourth, the RT cue was activated randomly (0–1.5 seconds window after the laser was broken) to avoid anticipation of the stimulus, but since older adults walked slower on average, the RT cue would be more likely to light up when they were farther from the obstacle. The farther distance provides more space and time to complete the RT task and to plan the obstacle crossing. Despite this apparent advantage, older adults were still at greater risk of lead limb contacts, and this effect would likely be further exaggerated if the timing was adjusted to be constant regardless of gait speed.

## Conclusion

Sensory interference was responsible for performance impairment in both younger and older adults, emphasizing the critical role of vision while obstacle crossing [9, 11, 45, 46, 61]. The

conclusion regarding the role of sensory interference is strengthened because performance was impaired with a discrete task that has low cognitive demand, and because changes in RT were only observed for the gait task, not the standing task. Gait behavior was affected differently for younger versus older adults: Changes in foot placement before the obstacle increased the risk of tripping with the lead foot in older adults, but increased risk of tripping with the trail foot in younger adults. Similarly, the proportion of lead versus trail limb obstacle contacts was higher for older adults. The increase in lead limb trip risk is relevant due to the shorter time for corrective action following a lead limb trip, which impairs the likelihood of recovering balance [38, 41, 62, 90].

## Author Contributions

**Conceptualization:** HyeYoung Cho, Shirley Rietdyk.

**Data curation:** HyeYoung Cho, Shirley Rietdyk.

**Formal analysis:** HyeYoung Cho.

**Investigation:** HyeYoung Cho, Shirley Rietdyk.

**Methodology:** HyeYoung Cho, Shirley Rietdyk.

**Project administration:** HyeYoung Cho.

**Resources:** Shirley Rietdyk.

**Software:** HyeYoung Cho.

**Supervision:** HyeYoung Cho, Shirley Rietdyk.

**Writing – original draft:** HyeYoung Cho, Shirley Rietdyk.

**Writing – review & editing:** HyeYoung Cho, Shirley Rietdyk.

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
