## [Decision Letter · Decision Letter 0]

3 Jan 2024

PONE-D-23-31634The effect of visual sensory interference during multitask obstacle crossing in younger and older adultsPLOS ONE

Dear Dr. Rietdyk,

Thank you for submitting your manuscript to PLOS ONE. After careful consideration, we feel that it has merit but does not fully meet PLOS ONE’s publication criteria as it currently stands. Therefore, we invite you to submit a revised version of the manuscript that addresses the points raised during the review process.

We look forward to receiving your revised manuscript.

Kind regards,

Sergiy Yakovenko

Academic Editor

PLOS ONE

2. We noted in your submission details that a portion of your manuscript may have been presented or published elsewhere. [Yes

The results from younger adult participants were previously published (Cho et al., 2019), and are included in the current manuscript as the comparison group for older adults.

Cho H, Romine NL, Barbieri FA, Rietdyk S. Gaze diversion affects cognitive and motor performance in young adults when stepping over obstacles. Gait Posture. 2019;73: 273–278. doi:10.1016/j.gaitpost.2019.07.380]

Please clarify whether this publication was peer-reviewed and formally published. If this work was previously peer-reviewed and published, in the cover letter please provide the reason that this work does not constitute dual publication and should be included in the current manuscript.

Additional Editor Comments:

Thank you for the high quality submission. Both reviewers have indicated only minor concerns/comments. Please address these requests in your resubmission. I would also like to add an additional Specific Comment:

Ln. 257 “... (speed was not different for high versus low sensory interference)... ” There might be an error here since Fig.5 indicates significant differences related to the sensory interference within young adults and elderly. Moreover, the positive correlation between the speed and interference magnitudes is unexpected within each group. Please discuss this result.

Reviewers' comments:

Reviewer's Responses to Questions

**Comments to the Author**

1. Is the manuscript technically sound, and do the data support the conclusions?

Reviewer #1: Yes

Reviewer #2: Yes

2. Has the statistical analysis been performed appropriately and rigorously? 

Reviewer #1: Yes

Reviewer #2: Yes

3. Have the authors made all data underlying the findings in their manuscript fully available?

Reviewer #1: Yes

Reviewer #2: Yes

4. Is the manuscript presented in an intelligible fashion and written in standard English?

Reviewer #1: Yes

Reviewer #2: Yes

5. Review Comments to the Author

Reviewer #1: Review of PLOS One manuscript: PONE-D-23-31634

Title: “The effect of visual sensory interference during multitask obstacle crossing in younger and older adults”

November 1, 2023

General comments:

The purpose of this research is to sensory interference alters performance under low levels of cognitive, temporal, and gait demand as well as to determine the effect of age on performance. They also examined learning effect as two blocks of tasks were performed. The study is well-done. The writing style is articulate and flows well. The introduction is in-depth and explains what other studies have done and how this work differs from them. Figure 1 is particularly informative. The methods are clear and provide sufficient detail to allow replication. The statistics are appropriate. The results are well-organized. The discussion is thorough, insightful, and includes the limitations of the study.

Specific comments:

Introduction: Figure 2 is informative and useful. It is unusual to have such a figure in the introduction rather than in the methods, but I can see how it provides information to clarify your purpose.

Methods: because obstacle crossing in older adults was assessed, did participants wear a harness to prevent a fall?

Methods: Did you control for shoes? Was the light in the room normally-lit by fluorescent lighting? I’m wondering if ambient light would be a factor.

Discussion, line 293: If not in this study, cite the study to which you refer: “In a different gait task, locomotor targeting, sensory interference….”

Discussion, line 392, in this usage, should multitask be multitasking? It seems to be a verb here, whereas in other instances you use it as a noun.

Reviewer #2: Preliminary Remarks:

The reviewer appreciates the authors' efforts in composing this paper and feels privileged to have the opportunity to review the manuscript. The topic is found to be quite interesting, and the paper is well-written. It is hoped that these comments will contribute to enhancing the quality of the paper before its publication.

Summary:

The study investigates the mutual interference between cognitive and locomotor tasks, specifically the dual-task effects, on an obstacle crossing task in both young and elderly individuals. In one condition, the light cue for a reaction time task was positioned on an obstacle to be crossed. In another condition, this cue was placed away from the obstacle, preventing simultaneous foveation of both the cue and the obstacle. Reaction time was influenced by the sensory interference condition during the obstacle crossing task, but not while standing. Notably, the effect of sensory interference on foot placement varied with age. The authors suggest that the increased distance before the obstacle observed in the elderly during the dual-task condition may relate to their diminished balance recovery capabilities.

Strength

The hypotheses are well-rationalized and clearly stated, facilitating easy comprehension of the methods, results, and discussion.

The use of an alpha level of 0.01 is commendable for addressing the issue of multiple comparisons.

Major comment

The term "sensory interference," as introduced in the second paragraph of the Introduction, is somewhat broad. While some readers might agree with this classification of the task condition, others could perceive it differently (e.g., as reflecting peripheral/focal vision). Therefore, the reviewer suggests using a more specific term (e.g., RT cue location) when describing the methods and results, to provide clearer and more universally understood what was done in the experiment and how the results were discussed.

Minor comments

Line 137: Please provide more detailed characteristics of the elderly participants, including their recruitment source. Were they independently living, community-dwelling, or residing in care homes?

Line 160: Was there any practice session for the RT task in the standing condition or for the obstacle crossing (walking) task? Clarification is needed regarding whether practice was allowed for dual-tasking.

Line 383: The discussion might benefit from considering the relatively low risk of the task as a factor explaining the absence of effect on clearance. Given that the obstacle was designed to collapse upon contact, did this reduce the participants' concern about clearance?

6. PLOS authors have the option to publish the peer review history of their article (what does this mean?). If published, this will include your full peer review and any attached files.

Reviewer #1: No

Reviewer #2: **Yes: **Masahiro Shinya

---

## [Author Response · Author response to Decision Letter 0]

2 Feb 2024

We thank the reviewers for their time and expertise in completing this review. We have addressed all the comments as outlined below. We believe that addressing the comments has improved the clarity and contribution of this manuscript. 

We noted in your submission details that a portion of your manuscript may have been presented or published elsewhere. [Yes, The results from younger adult participants were previously published (Cho et al., 2019), and are included in the current manuscript as the comparison group for older adults.

Cho H, Romine NL, Barbieri FA, Rietdyk S. Gaze diversion affects cognitive and motor performance in young adults when stepping over obstacles. Gait Posture. 2019;73: 273–278. doi:10.1016/j.gaitpost.2019.07.380]

Please clarify whether this publication was peer-reviewed and formally published. If this work was previously peer-reviewed and published, in the cover letter please provide the reason that this work does not constitute dual publication and should be included in the current manuscript.

RESPONSE: We apologize for the confusion. Here is a clear description (text also provided in cover letter):

• Previous manuscript: Total of 17 younger adult participants. Peer-reviewed and formally published (Cho et al., 2019). 

• Current manuscript: Total of 31 participants, 17 younger adults and 14 older adults. While the 17 younger adults were published previously, the results for the 14 older adults have not been published elsewhere. The 17 younger adults are used as a comparison group only. 

• Therefore, the work does not constitute dual publication. Text has been added to the manuscript for clarity (Line 136).

Additional Editor Comments:

Ln. 257 “... (speed was not different for high versus low sensory interference)... ” There might be an error here since Fig.5 indicates significant differences related to the sensory interference within young adults and elderly. Moreover, the positive correlation between the speed and interference magnitudes is unexpected within each group. Please discuss this result.

RESPONSE: We apologize for the confusion. For the comment regarding the possible error in Fig. 5: While there are three levels of sensory interference, only the baseline level was different from the other two levels (high and low sensory interference). We have edited the text to increase clarity. (Lines 263-267)

For the comment regarding the positive correlation between the speed and interference magnitudes within each group, I believe you are referring to the unexpected faster speed when sensory interference was in place (both high and low sensory interference) versus the baseline condition. We discuss this in the discussion section, lines: 407-415 

 

Reviewer #1

General comments: The purpose of this research is to sensory interference alters performance under low levels of cognitive, temporal, and gait demand as well as to determine the effect of age on performance. They also examined learning effect as two blocks of tasks were performed. The study is well-done. The writing style is articulate and flows well. The introduction is in-depth and explains what other studies have done and how this work differs from them. Figure 1 is particularly informative. The methods are clear and provide sufficient detail to allow replication. The statistics are appropriate. The results are well-organized. The discussion is thorough, insightful, and includes the limitations of the study.

Specific comments: Introduction: Figure 2 is informative and useful. It is unusual to have such a figure in the introduction rather than in the methods, but I can see how it provides information to clarify your purpose.

RESPONSE: Thank you for your positive comments. 

Methods: because obstacle crossing in older adults was assessed, did participants wear a harness to prevent a fall?

RESPONSE: Participants did not wear a harness. We have added the following text to the methods to increase clarity. “The obstacle was designed to fall forward if contacted, like a hurdle, to decrease fall-risk. Thus, participants were not harnessed. No one fell in this protocol or in our previous studies with this obstacle design (e.g., Muir et al., 2015; Becker & Rietdyk, 2022).” (lines 154-157)

Methods: Did you control for shoes? Was the light in the room normally-lit by fluorescent lighting? I’m wondering if ambient light would be a factor.

RESPONSE: We have added text regarding shoes to increase clarity. “Participants wore comfortable walking or athletic shoes; they were asked to wear the shoes they habitually wear when walking.” See lines 146-147. Each subject wore their own shoes, thus, within a subject, the shoes were held constant for all conditions. The room was normally-lit; the lighting was constant for all subjects and for all conditions. Therefore, we believe that shoes and lighting did not affect changes in behavior due to manipulations of sensory interference. 

Discussion, line 293: If not in this study, cite the study to which you refer: “In a different gait task, locomotor targeting, sensory interference….”

RESPONSE: Changes made as requested (reference was Berg and Murdock 2011). See line 304

Discussion, line 392, in this usage, should multitask be multitasking? It seems to be a verb here, whereas in other instances you use it as a noun.

RESPONSE: Changes made as requested. See line 407 

Reviewer #2: Preliminary Remarks:

The reviewer appreciates the authors' efforts in composing this paper and feels privileged to have the opportunity to review the manuscript. The topic is found to be quite interesting, and the paper is well-written. It is hoped that these comments will contribute to enhancing the quality of the paper before its publication.

Summary:

The study investigates the mutual interference between cognitive and locomotor tasks, specifically the dual-task effects, on an obstacle crossing task in both young and elderly individuals. In one condition, the light cue for a reaction time task was positioned on an obstacle to be crossed. In another condition, this cue was placed away from the obstacle, preventing simultaneous foveation of both the cue and the obstacle. Reaction time was influenced by the sensory interference condition during the obstacle crossing task, but not while standing. Notably, the effect of sensory interference on foot placement varied with age. The authors suggest that the increased distance before the obstacle observed in the elderly during the dual-task condition may relate to their diminished balance recovery capabilities.

Strength

The hypotheses are well-rationalized and clearly stated, facilitating easy comprehension of the methods, results, and discussion.

The use of an alpha level of 0.01 is commendable for addressing the issue of multiple comparisons.

RESPONSE: Thank you for your positive comments.

Major comment

The term "sensory interference," as introduced in the second paragraph of the Introduction, is somewhat broad. While some readers might agree with this classification of the task condition, others could perceive it differently (e.g., as reflecting peripheral/focal vision). Therefore, the reviewer suggests using a more specific term (e.g., RT cue location) when describing the methods and results, to provide clearer and more universally understood what was done in the experiment and how the results were discussed.

RESPONSE: We agree that what we manipulated was the RT cue location, and we assumed that this manipulation resulted in high vs low sensory interference. However, when naming the condition, we were left with long labels: “RT cue on obstacle”, and “RT cue away from obstacle” which reduced clarity in the results and in the figure legends (it was originally written in this manner, and after feedback, we decided to refer to them more simply as high vs low sensory interference). Further, the reader would not necessarily associate the condition label with the intention (high vs low sensory interference). Thus, we opted to keep the sensory interference label, but have added text to highlight our assumption. (lines 161-163 and lines 467-475)

Minor comments

Line 137: Please provide more detailed characteristics of the elderly participants, including their recruitment source. Were they independently living, community-dwelling, or residing in care homes? 

RESPONSE: “The older adults lived independently (i.e., they did not require assistance with daily activities).” We have added the text to increase clarity. (Lines 138-139)

Line 160: Was there any practice session for the RT task in the standing condition or for the obstacle crossing (walking) task? Clarification is needed regarding whether practice was allowed for dual-tasking. 

RESPONSE: Practice sessions for the RT task were in the standing condition. We have edited the text to increase clarity. (Line 168)

Line 383: The discussion might benefit from considering the relatively low risk of the task as a factor explaining the absence of effect on clearance. Given that the obstacle was designed to collapse upon contact, did this reduce the participants' concern about clearance?

RESPONSE: That is an interesting observation. We have added text to address this idea. “An alternate explanation is that the low-risk/collapsible obstacle resulted in complacency during this lab task and participants did not adjust their foot clearance. However, it seems unlikely that complacency would not affect foot clearance but would affect foot placement. In addition, in previous studies with collapsible obstacles, foot clearance has been altered by visual manipulations (Rhea & Rietdyk, 2011; Rietdyk & Rhea, 2011).” (Lines 399-403)

---

## [Decision Letter · Decision Letter 1]

15 Apr 2024

The effect of visual sensory interference during multitask obstacle crossing in younger and older adults

PONE-D-23-31634R1

Dear Dr. Rietdyk,

We’re pleased to inform you that your manuscript has been judged scientifically suitable for publication and will be formally accepted for publication once it meets all outstanding technical requirements.

Kind regards,

Sergiy Yakovenko

Academic Editor

PLOS ONE

Additional Editor Comments (optional):

Thank you for the high-quality submission to PLoS One.

Reviewers' comments:

Reviewer's Responses to Questions

**Comments to the Author**

1. If the authors have adequately addressed your comments raised in a previous round of review and you feel that this manuscript is now acceptable for publication, you may indicate that here to bypass the “Comments to the Author” section, enter your conflict of interest statement in the “Confidential to Editor” section, and submit your "Accept" recommendation.

Reviewer #1: All comments have been addressed

Reviewer #2: All comments have been addressed

2. Is the manuscript technically sound, and do the data support the conclusions?

Reviewer #1: Yes

Reviewer #2: Yes

3. Has the statistical analysis been performed appropriately and rigorously? 

Reviewer #1: Yes

Reviewer #2: Yes

4. Have the authors made all data underlying the findings in their manuscript fully available?

Reviewer #1: Yes

Reviewer #2: (No Response)

5. Is the manuscript presented in an intelligible fashion and written in standard English?

Reviewer #1: Yes

Reviewer #2: (No Response)

6. Review Comments to the Author

Reviewer #1: The authors have addressed my prior concerns and I have no further suggestions or questions on this paper.

Reviewer #2: (No Response)

7. PLOS authors have the option to publish the peer review history of their article (what does this mean?). If published, this will include your full peer review and any attached files.

Reviewer #1: **Yes: **Jean L. McCrory

Reviewer #2: No

---

## [Editor Report · Acceptance letter]

29 Apr 2024

PONE-D-23-31634R1 

PLOS ONE

Dear Dr. Rietdyk, 

I'm pleased to inform you that your manuscript has been deemed suitable for publication in PLOS ONE. Congratulations! Your manuscript is now being handed over to our production team.

Kind regards, 

on behalf of

Dr. Sergiy Yakovenko 

Academic Editor

PLOS ONE